# Exemplar Guided Unsupervised Image-to-Image Translation with Semantic Consistency

**Liqian Ma**[1]    **Xu Jia**[4*]    **Stamatios Georgoulis**[1,3]    **Tinne Tuytelaars**[2]    **Luc Van Gool**[1,3]
[1]KU-Leuven/PSI, TRACE (Toyota Res in Europe)    [2]KU-Leuven/PSI    [3]ETH Zurich
[4]Huawei Noah's Ark Lab
{liqian.ma, xu.jia, tinne.tuytelaars, luc.vangool}@esat.kuleuven.be
{georgous, vangool}@vision.ee.ethz.ch

## Abstract

Image-to-image translation has recently received significant attention due to advances in deep learning. Most works focus on learning either a one-to-one mapping in an unsupervised way or a many-to-many mapping in a supervised way. However, a more practical setting is many-to-many mapping in an unsupervised way, which is harder due to the lack of supervision and the complex inner- and cross-domain variations. To alleviate these issues, we propose the Exemplar Guided & Semantically Consistent Image-to-image Translation (EGSC-IT) network which conditions the translation process on an exemplar image in the target domain. We assume that an image comprises of a content component which is shared across domains, and a style component specific to each domain. Under the guidance of an exemplar from the target domain we apply Adaptive Instance Normalization to the shared content component, which allows us to transfer the style information of the target domain to the source domain. To avoid semantic inconsistencies during translation that naturally appear due to the large inner- and cross-domain variations, we introduce the concept of feature masks that provide coarse semantic guidance without requiring the use of any semantic labels. Experimental results on various datasets show that EGSC-IT does not only translate the source image to diverse instances in the target domain, but also preserves the semantic consistency during the process.

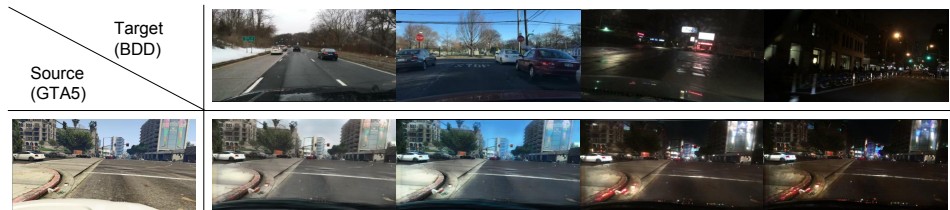

Figure 1: Exemplar guided image translation examples of GTA5 → BDD. Best viewed in color.

## 1 Introduction

Image-to-image (I2I) translation refers to the task of mapping an image from a source domain to a target domain, *e.g.* semantic maps to real images, gray-scale to color images, low-resolution to high-resolution images, and so on. The recent advances in deep learning have greatly improved the quality of I2I translation methods for a number of applications, including super-resolution (Dong et al., 2014), colorization (Zhang et al., 2016), inpainting (Pathak et al., 2016), attribute transfer (Lee et al., 2018), style transfer (Gatys et al., 2016), and domain adaptation (Hoffman et al., 2018; Liu et al., 2017). Most of these works (Isola et al., 2017; Wang et al., 2018; Zhu et al., 2017b) have been very successful in these cross-domain I2I translation tasks because they rely on large datasets of paired training data as supervision. However, for many tasks it is not easy, or even possible, to obtain such paired data that show how an image in the source domain should be translated to an image in the target domain, *e.g.* in cross-city street view translation or male-female face translation. For this

---

*Part of this work was done when Xu Jia was in KU Leuven

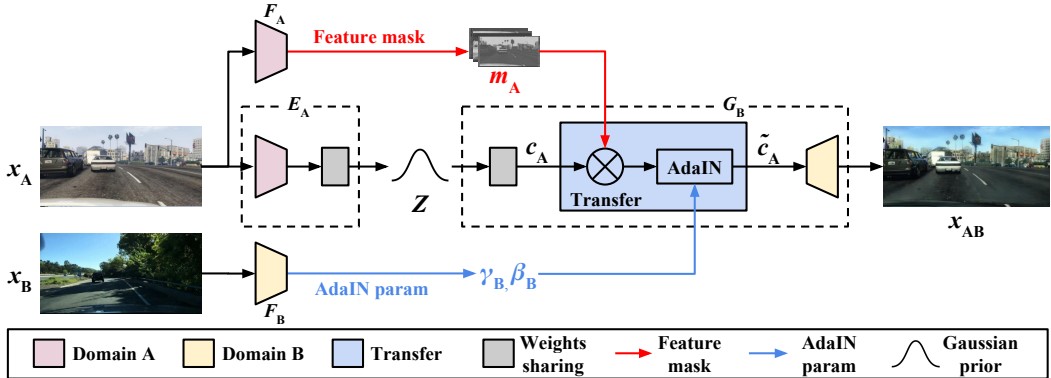

Figure 2: The $x_A$ to $x_{AB}$ translation procedure of our EGSC-IT framework. 1) Source domain image $x_A$ is fed into an encoder $E_A$ to compute a shared latent code $z_A$ and is further decoded to a common high-level content representation $c_A$. 2) Meanwhile, $x_A$ is also fed into a sub-network to compute feature masks $m_A$. 3) The target domain exemplar image $x_B$ is fed to a sub-network to compute affine parameters $\gamma_B$ and $\beta_B$ for AdaIN . 4) The content representation $c_A$ is transferred to the target domain using $m_A$, $\gamma_B$, $\beta_B$, and is further decoded to an image $x_{AB}$ by target domain generator $G_B$.

unsupervised setting, Zhu et al. (2017a) proposed to use a cycle-consistency loss, which assumes that a mapping from domain A to B, followed by its reverse operation approximately yields an identity function, that is, $F(G(x_A)) \approx x_A$. Liu et al. (2017) further proposed a shared-latent space constraint, which assumes that a pair of corresponding images $(x_A, x_B)$ from domains A and B respectively can be mapped to the same representation $z$ in a shared latent space Z. Note that, all the aforementioned methods assume that there is a deterministic one-to-one mapping between the two domains, *i.e.* each image in A is translated to only a single image in B. By doing so, they fail to capture the multimodal nature of the image distribution within the target domain, *e.g.* different color and style of shoes in sketch-to-image translation and different seasons in synthetic-to-real street view translation.

In this work, we propose Exemplar Guided & Semantically Consistent I2I Translation (EGSC-IT) to explicitly address this issue. As shown in concurrent works (Gonzalez-Garcia et al., 2018; Huang et al., 2018; Lee et al., 2018), we assume that an image is composed of two disentangled representations. In our case, first a domain-shared representation that models the content in the image, and second a domain-specific representation that contains the style information. However, for a multimodal domain with complex inner-variations, as the ones we target in this paper, *e.g.* street views of day-and-night or different seasons, it is difficult to have a single static representation which covers all variations in that domain. Moreover, it is unclear which style (time-of-day/season) to pick during the image translation process. To handle such multimodal I2I translations, some approaches (Almahairi et al., 2018; Gonzalez-Garcia et al., 2018; Lee et al., 2018) incorporate noise vectors as additional inputs to the generator, but as shown in (Isola et al., 2017; Zhu et al., 2017b) this could lead to mode collapsing issues. Instead, we propose to condition the image translation process on an arbitrary image in the target domain, *i.e.* an exemplar. By doing so, EGSC-IT does not only enable multimodal (*i.e.* many-to-many) image translations, but also allows for explicit control over the translation process, since by using different exemplars as guidance we are able to translate an input image into images of different styles within the target domain – see Fig. 1.

To instantiate this idea, we adopt the weight sharing architecture proposed in UNIT (Liu et al., 2017), but instead of having a single latent space shared by both domains, we propose to decompose the latent space into two components according to the two disentangled representations presented above. That is, a domain-shared component that focuses on the image content, and a domain-specific component that captures the style information associated with the exemplar. In our particular case, the domain-shared content component contains semantic information, such as the objects' category, shape and spatial layout, while the domain-specific style component contains the style information, such as the color and texture, to be translated from a target domain exemplar to an image in the source domain. To realize this translation, we apply adaptive instance normalization (AdaIN) (Huang & Belongie, 2017) to the shared content component of the source domain image using the AdaIN parameters computed from the target domain exemplar. However, directly applying AdaIN to the feature maps of the shared content component would mix up all objects and scenes in the image,

making the image translation prone to failure when an image contains diverse objects and scenes. To tackle this problem, existing works (Gatys et al., 2017; Hoffman et al., 2018; Li et al., 2018; Luan et al., 2017) use semantic labels as an additional form of supervision. However, ground-truth semantic labels are not easy to obtain for most tasks as they require labor-intensive annotations. Instead, to maintain the semantic consistency during image translation without using any semantic labels we propose to compute feature masks. One can think of feature masks as attention modules that approximately decouple different semantic categories in an unsupervised way under the guidance of perceptual losses and adversarial loss. In particular, one feature mask corresponding to a certain semantic category is applied to one feature map of the shared content component, and consequently the AdaIN for that channel is only required to capture and model the style difference for that category, *e.g.* sky's style in two domains. To the best of our knowledge, this is the first line of work that addresses the semantic consistency issue under this setting. See Fig. 2 for an overview of EGSC-IT.

Our contribution is three-fold. i) We propose a novel approach for the I2I translation task, which enables multimodal (*i.e.* many-to-many) mappings and allows for explicit style control over the translation process. ii) We introduce the concept of feature masks for the unsupervised, multimodal I2I translation task, which provides coarse semantic guidance without using any semantic labels. iii) Evaluation on different datasets show that our method is robust to mode collapse and can generate results with semantic consistency, conditioned on a given exemplar image.

## 2 RELATED WORK

**I2I translation.** I2I translation is used to learn a mapping from one image (*i.e.* source domain) to another (*i.e.* target domain). Recently, with the advent of generative models (Goodfellow et al., 2014; Kingma & Welling, 2013), there have been a lot of works on this topic. Isola et al. (2017) proposed pix2pix to learn the mapping from input images to output images using a U-Net neural network in an adversarial way. Wang et al. (2018) extended the method to pix2pixHD, to turn semantic label maps into high-resolution photo-realistic images. Zhu et al. (2017b) extended pix2pix to BicycleGAN, which can model multimodal distributions and produce both diverse and realistic results. All these methods, however, require paired training data as supervision which may be difficult or even impossible to collect in many scenarios, such as synthetic-to-real street view translation or face-to-cartoon translation (Royer et al., 2017).

Recently, several unsupervised methods have been proposed to learn the mappings between two image collections without paired training data. Note that, this is an ill-posed problem since there are infinitely many mappings existing between two unpaired image domains. To address this ill-posed problem, different constraints have been added to the network to regularize the learning process (Kim et al., 2017; Liu et al., 2017; Royer et al., 2017; Yi et al., 2017; Zhu et al., 2017a). One popular constraint is cycle-consistency, which enforces the network to learn deterministic mappings for various applications. Going one step further, Liu et al. (2017) proposed a shared-latent space constraint which encourages a pair of images from different domains to be mapped to the same representation in the latent space. In a similar vein, Royer et al. (2017) proposed to enforce a feature-level constraint with a latent embedding reconstruction loss. However, we argue that these constraints are not well suited for complex domains with large inner-domain variations, as also mentioned in (Almahairi et al., 2018; Gonzalez-Garcia et al., 2018; Lee et al., 2018; Lin et al., 2018). Unlike these methods, to address this problem we propose to add a target domain exemplar as guidance during image translation through AdaIN (Huang & Belongie, 2017). As explained in the previous section, the AdaIN technique is utilized to transfer the style component from the target domain exemplar to the shared content component of the source domain image. This allows multimodal (*i.e.* many-to-many) translations and can produce images of desired styles with explicit control over the translation process. Concurrent to our work, MUNIT (Huang et al., 2018), also proposed to use AdaIN to transfer style information from the target domain to the source domain. Unlike MUNIT, before applying AdaIN to the shared content component we compute feature masks to decouple different semantic categories and preserve the semantic consistency during the translation process. In particular, by applying feature masks to the feature maps of the shared content component, each channel can specialize and model the style difference only for a single semantic category, which is crucial when handling domains with complex scenes.

**Style transfer.** Style transfer aims at transferring the style information from an exemplar image to a content image, while preserving the content information. The seminal work by Gatys et al. (2016)

Table 1: Comparison of unpaired I2I translation networks: CycleGAN (Zhu et al., 2017a), UNIT (Liu et al., 2017), Augmented CycleGAN (Almahairi et al., 2018), CDD-IT (Gonzalez-Garcia et al., 2018), DRIT (Lee et al., 2018), MUNIT (Huang et al., 2018), EGSC-IT (Ours).

| | Multi modal | Disentangle & InfoFusion | Feature mask | Semantic consistency | Perceptual loss |
|---|---|---|---|---|---|
| CycleGAN | - | - | - | Low | - |
| UNIT | - | - | - | Low | - |
| Augmented CycleGAN | ✓ | - | - | Low | - |
| CDD-IT | ✓ | Swap feature | - | Low | - |
| DRIT | ✓ | Swap feature | - | Low | - |
| MUNIT | ✓ | AdaIN | - | Middle | Depends |
| EGSC-IT (Ours) | ✓ | AdaIN | ✓ | High | ✓ |

proposed to transfer style information by matching the feature correlations, *i.e.* Gram matrices, in the convolutional layers of a deep neural network (DNN) following an iterative optimization process. In order to improve the speed and flexibility, several feed-forward neural networks have been proposed. Huang & Belongie (2017) proposed a simple but effective method, called AdaIN, which aligns the mean and variance of the content image features with those of the style image features. Li et al. (2017) proposed the whitening and coloring transform (WCT) algorithm, which directly matches the features' covariance in the content image to those in the given style image. However, due to the lack of semantic consistency during translation, these stylization methods usually generate non-photorealistic images, suffering from the "spills over" problem (Luan et al., 2017). To address this, semantic label maps are used as additional supervision to help style transfer between corresponding semantic regions (Gatys et al., 2017; Li et al., 2018; Luan et al., 2017). Unlike these works, we propose to compute feature masks to approximately model such semantic information without using any semantic labels that are very hard to collect.

Table 1 summarizes the features of the most related works. As can be seen, our method using the combination of AdaIN and feature masks under the guidance of perceptual loss is, to the best of our knowledge, the first to achieve multimodal I2I translations in the unsupervised setting with *high semantic consistency*, without requiring any ground-truth semantic labels.

## 3 METHOD

Our goal is to learn a many-to-many mapping between two domains in an unsupervised way, which is guided by the style of an exemplar while retaining the semantic consistency at the same time. For example, a synthetic street view image can be translated to either a day-time or night-time realistic scene, depending on the exemplar. To realize this, similarly to concurrent works (Gonzalez-Garcia et al., 2018; Huang et al., 2018; Lee et al., 2018) we assume that an image can be decomposed into two disentangled components. In our case, that is, one modeling the shared content between domains, *i.e.* domain-shared content component, and another modeling the style information specific to exemplars in the target domain, *i.e.* domain-specific style component. In what follows, we present our EGSC-IT framework, the architecture of its networks, and the learning procedure.

### 3.1 FRAMEWORK

For simplicity, we present EGSC-IT in the A→B direction – see Fig. 2. Each image domain (*i.e.* source and target) is modeled by a VAE-GAN Larsen et al. (2016), which includes an encoder $E_A$, a generator $G_A$, and a discriminator $D_A$. For the B→A direction, the translation process as well as the notation are analogous.

**Weight sharing for domain-shared content.** To learn the content component of an image pair that is shared across source and target domains we employ the weight sharing strategy proposed in UNIT (Liu et al., 2017). The latter assumes that the two domains, A and B, share a common latent space, and any image pair from the two domains, $x_A$ and $x_B$, can be mapped to the same latent representation in this shared-latent space $z$. They achieve this by simply sharing the weights of the last layer in $E_A$ and $E_B$ as well as the first layer in $G_A$ and $G_B$. For more details about the weight-sharing strategy we refer the reader to the original UNIT paper.

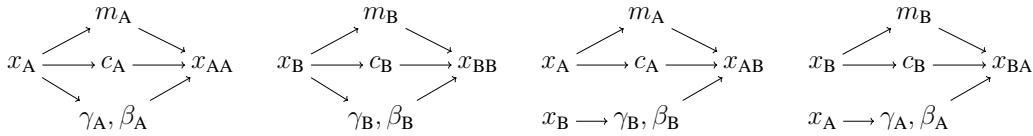

Figure 3: Information flow diagrams of auto-encoding procedures $x_A \rightarrow x_{AA}$ and $x_B \rightarrow x_{BB}$, and translation procedures $x_A \rightarrow x_{AB}$ and $x_B \rightarrow x_{BA}$.

**Exemplar-based AdaIN for domain-specific style.** The shared content component contains semantic information, such as the objects' category, shape and spatial layout, but no style information, *e.g.* their color and texture. Inspired by Huang & Belongie (2017), who showed that AdaIN's affine parameters have a big influence on the output image's style, we propose to apply AdaIN to the shared content component before the decoding stage. In particular, the exemplar from the target domain is fed to another network (see Fig. 2, blue line) to compute a set of feature maps $f_B$, which are expected to contain the style information of the target domain. As in (Huang & Belongie, 2017), means and variances are calculated for each channel of $f_B$ and used as AdaIN's affine parameters,

$$\gamma_B = \delta(f_B), \quad \beta_B = \mu(f_B), \tag{1}$$

$$\text{AdaIN}(c_A, \gamma_B, \beta_B) = \gamma_B \frac{(c_A - \mu(c_A))}{\delta(c_A)} + \beta_B, \tag{2}$$

where $\mu(\cdot)$ and $\delta(\cdot)$ respectively denote a function to compute the mean and variance across spatial dimensions. The shared content component is first normalized by these affine parameters, as shown in Eq. 2, and then decoded to a target-domain image using the target domain generator. Since different affine parameters normalize the feature statistics in different ways, by using different exemplar images in the target domain as input we can translate an image in the source domain to different sub-styles in the target domain. Therefore, EGSC-IT does not only allow for multimodal I2I translations, but at the same time enables the user to have explicit style control over the translation process.

**Feature masks for semantic consistency.** Directly applying AdaIN to the shared content component does not give satisfying results. The reason is that one channel in the shared content component is likely to contain information from multiple objects and scenes. The difference of these objects and scenes between the two domains is not always uniform, due to the large inner- and cross-domain variations. As such, applying AdaIN over a feature map with complex semantics is prone to mix styles of different objects and scenes together, hence failing to give semantically-consistent translations. To tackle this problem, existing works use semantic labels as an additional form of supervision. However, ground-truth semantic labels are not easy to obtain for most tasks as they require labor-intensive annotations. Instead, we propose to compute feature masks (see Fig. 2, red line) to make an approximate estimation of semantic categories without using any ground-truth semantic labels. The feature masks $m_A$, which can be regarded as attention modules, are computed by applying a nonlinear activation function and a threshold to feature maps $f_A$, *i.e.* $m_A = (1 - \eta) \cdot \sigma(f_A) + \eta$, where $\eta$ is a threshold and $\sigma$ is the sigmoid function. Feature masks contain substantial semantic information, which can be used to retain the semantic consistency during translation, *e.g.* translating the source sky into the target sky style without affecting the other scene elements. The new normalized representation $\tilde{c}_A$ is $\tilde{c}_A = \text{AdaIN}(m_A \circ c_A, \gamma_B, \beta_B)$, where $\circ$ denotes the Hadamard product.

During training, there are four types of information flow – see Fig. 3. For the reconstruction flow $x_A \rightarrow x_{AA}$, the shared content component $c_A$, feature masks $m_A$, and AdaIN parameters $\gamma_A, \beta_A$ are all computed from $x_A$ (and vice versa for $x_B \rightarrow x_{BB}$). For the translation flow $x_A \rightarrow x_{AB}$, the shared content component $c_A$ and feature masks are computed from $x_A$, while AdaIN's affine parameters $\gamma_B$ and $\beta_B$ are computed from the target domain exemplar $x_B$ (and vice versa for $x_B \rightarrow x_{BA}$).

## 3.2 NETWORK ARCHITECTURE

The overall framework can be divided into several sub-networks[1]. 1) Two Encoders, $E_A$ and $E_B$. Each one consists of several strided convolutional layers and several residual blocks to compute the shared content component. 2) A feature mask network and an AdaIN network, $F_A$ and $F_B$ for A $\rightarrow$ B translation (vise versa for B $\rightarrow$ A) have the same architecture as the Encoder above except

---

[1]For more details we refer the reader to the supplementary material.

for the weight-sharing layers. 3) Two Generators, $G_A$ and $G_B$, are almost symmetric to the Encoders except that the up-sampling is done by transposed convolutional layers. 4) Two Discriminators, $D_A$ and $D_B$, are fully-convolutional networks containing a stack of convolutional layers. 5) A VGG sub-network (Simonyan & Zisserman, 2015), $VGG$, that contains the first few layers (up to relu5_1) of a pre-trained VGG-19 (Simonyan & Zisserman, 2015), which is used to calculate perceptual losses. Note that, although we use UNIT as our baseline framework to build upon, this is not a hard restriction. In theory, UNIT can be replaced with any baseline framework with similar functionality.

## 3.3 LEARNING

The learning procedure of EGSC-IT contains VAEs, GANs, cycle-consistency and perceptual losses. To make the training more stable, we first pre-train the feature mask network and AdaIN network for each domain separately within a VAE-GAN architecture, and use the encoder part as fixed feature extractors, *i.e.* $F_A$ and $F_B$, for the remaining training. The overall loss is shown in Eq. 3,

$$
\begin{aligned}
\mathcal{L}(E_A, G_A, D_A, E_B, G_B, D_B) =& \mathcal{L}_{\text{VAE}_A}(E_A, G_A) + \mathcal{L}_{\text{GAN}_A}(E_A, G_A, D_A) + \mathcal{L}_{\text{CC}_A}(E_A, G_A, E_B, G_B) \\
&+ \mathcal{L}_{\text{VAE}_B}(E_B, G_B) + \mathcal{L}_{\text{GAN}_B}(E_B, G_B, D_B) + \mathcal{L}_{\text{CC}_B}(E_A, G_A, E_B, G_B) \\
&+ \mathcal{L}_P(E_A, G_A, E_B, G_B),
\end{aligned}
\tag{3}
$$

where the VAEs, GANs and cycle-consistency losses are identical to the ones used in Liu et al. (2017). The perceptual loss consists of the content loss captured by $VGG19$ feature maps $\phi$ containing localized spatial information, and the style loss captured by the Gram matrix containing non-localized style information similar to (Gatys et al., 2016; Johnson et al., 2016), is as follows,

$$
\mathcal{L}_P(E_A, G_A, E_B, G_B) = \lambda_c \mathcal{L}_{c_A}(E_A, G_A) + \lambda_s \mathcal{L}_{s_A}(E_A, G_A) + \lambda_c \mathcal{L}_{c_B}(E_B, G_B) + \lambda_s \mathcal{L}_{s_B}(E_B, G_B), \tag{4}
$$

where $\lambda_c$ and $\lambda_s$ are the weights for content and style losses, which depend on the dataset domain variations and tasks. The content loss $\mathcal{L}_{c_A}(E_A, G_A)$ and style loss $\mathcal{L}_{s_A}(E_A, G_A)$ are defined as,

$$
\mathcal{L}_{c_A}(E_A, G_A) = \mathbb{E}[\|\phi(x_{AB}) - \phi(x_A)\|_1], \mathcal{L}_{s_A}(E_A, G_A) = \mathbb{E}[\|Gram(x_{AB}) - Gram(x_B)\|_1], \tag{5}
$$

We use the first convolutional layer of the five blocks in $VGG19$ to extract the feature maps. $\mathcal{L}_{c_B}(E_B, G_B)$ and $\mathcal{L}_{s_B}(E_B, G_B)$ are defined likewise. For the content losses $\mathcal{L}_{c_A}$ and $\mathcal{L}_{c_B}$, a linear weighting scheme is adopted to help the network focus more on the high-level semantic information. In both content and style losses we use the L1 distance, which in our experiments outperforms L2.

Now that we have introduced all losses, we can explain how these losses help to achieve I2I translation, multimodal translation, and semantic consistency. *I2I translation*: $\mathcal{L}_{\text{VAE}}$, $\mathcal{L}_{\text{GAN}}$ and $\mathcal{L}_{\text{CC}}$ help to maintain the shared latent space by relating the two different domains and finding the optimal translation between the two in an unsupervised way. *Multimodal translation*: $\mathcal{L}_S$ and $\mathcal{L}_{\text{GAN}}$ help to encourage $x_{AB}$ to look not only like the main mode of variation in domain B, but also like an exemplar from domain B, $x_B$, since the domain space is actually supported by each data sample. *Semantic consistency*: $\mathcal{L}_C$ encourages the network to utilize the feature mask information for semantic consistency, without relying on hard correspondences between semantic labels as existing works do.

## 4 EXPERIMENTS

We evaluate EGSC-IT's translation ability, *i.e.* how well it generates domain-realistic-looking and semantically consistent images, both qualitatively and quantitatively on three tasks with progressively increasing visual complexity: 1) single-digit translation; 2) multi-digit translation; 3) street view translation. We first perform an ablation study on various components of EGSC-IT on the single-digit translation task. Then, we present results on more challenging translation tasks, and evaluate EGSC-IT quantitatively on the semantic segmentation task. In supplementary material, we apply EGSC-IT to the face gender translation task and perform the ablation study on the street-view translation task.

**Single-digit translation.** We set up a controlled experiment on the MNIST-Single dataset, which is created based on the handwritten digits dataset MNIST LeCun et al. (1998). The MNIST-Single dataset consists of two different domains as shown in Fig. 4. For domain A of both training/test sets, the foreground and background are randomly set to *black* or *white* but different from each other. For domain B of training set, the foreground and background for digits from 0 to 4 are randomly assigned

Figure 4: Single-digit translation testing results. The left-most four columns are samples from domain $x_A$ and $x_B$, and reference translated ground truth $x_{AB}$ and $x_{BA}$. * Models are trained using MNIST-Single data as EGSC-IT. Best viewed in color.

Table 2: SSIM evaluation for single-digit translation. Higher is better.

|  | CycleGAN | UNIT | MUNIT | EGSC-IT w/o feature mask | EGSC-IT w/o AdaIN | EGSC-IT w/o $\mathcal{L}_P$ | EGSC-IT |
|---|---|---|---|---|---|---|---|
| $A \to B$ | 0.214±0.168 | 0.178±0.160 | 0.463±0.094 | 0.395±0.137 | 0.208±0.166 | 0.286±0.183 | **0.478± 0.090** |
| $B \to A$ | 0.089±0.166 | 0.074±0.158 | 0.227±0.128 | 0.133±0.171 | 0.080±0.167 | 0.093±0.169 | **0.232± 0.131** |

a color from $\{red, green, blue\}$, and the foreground and background for digits from 5 to 9 are fixed to *red* and *green*, respectively. For domain B of testing set, the foreground and background of all digits are randomly assigned a color from $\{red, green, blue\}$. Such data imbalance is designed on purpose to test the translation diversity and generalization ability. In particular, for diversity, we want to check whether a method would suffer from the mode collapse issue and translate the images to the dominant mode, *i.e.* *(red, green)*, while for generalization, we want to check whether the model can be applied to new styles in the target domain that never appear in the training set, *e.g.* translate number 6 from *black* foreground and *white* background to *blue* foreground and *red* background.

We first analyze the importance of three main components of EGSC-IT, *i.e.* feature masks, AdaIN, and perceptual loss, on the MNIST-Single dataset. As shown in Fig. 4, EGSC-IT can successfully transfer the source image into the style of the exemplar image. Ablating the feature mask from EGSC-IT, leads to incorrect foreground and background shape, indicating that feature masks can indeed provide semantic information to transfer the corresponding local regions. Without AdaIN, the network suffers from the mode collapse issue in A→B translation, *i.e.* all samples are transferred to the dominant mode with $red$ foreground and $green$ background, indicating that the exemplar's style information can help the network to learn many-to-many mappings and avoid the mode collapse issue. Without perceptual losses $\mathcal{L}_P$, colors of foreground and background are incorrect, which shows that perceptual losses can encourage the network to learn semantic knowledge, in this case foreground and background, without ground-truth semantic labels. As for other I2I translation methods, CycleGAN (Zhu et al., 2017a) and UNIT (Liu et al., 2017) can only do deterministic image translations and suffer from mode collapse issue, such as *white* ↔ *green* and *black* ↔ *red* for CycleGAN in Fig. 4. MUNIT (Huang et al., 2018) can successfully transfer the style of the exemplar image, but the foreground and background are mixed and the digit's shape is not kept well. These qualitative observations are in accordance with the quantitative results in Tab. 2, where our full EGSC-IT obtains higher SSIM scores than all other alternatives. In addition, we compare with other style transfer methods, Neural ST (Gatys et al., 2016), AdaIN (Huang & Belongie, 2017), and WCT (Li et al., 2017). In each case, we resize the input image to $512 \times 512$ resolution and choose the best performing hyper-parameters. Note how style transfer methods can transfer the style successfully but fail to keep semantic consistency. Quantitative results for style transfer methods are in supplementary material.

To verify EGSC-IT's ability to match the target domain distributions of real data and translated results, we visualize them using t-SNE embeddings (Maaten & Hinton, 2008) in Fig. 5. The t-SNE embeddings are calculated from the translated images with PCA dimension reducing. Our method can match the distributions well, while others either collapse to few modes or mismatch the distributions.

**Multi-digit translation.** The MNIST-Multiple dataset is another controlled experiment designed to mimic the complexity in real-world scenarios. It is used to test whether the network understands the semantics, *i.e.* digits, in an image and translates each digit accordingly. Each image in MNIST-Multiple contains all ten digits, which are randomly placed in $4 \times 4$ grids. Two domains are designed: in domain A, the foreground and background are randomly set to *black* or *white*, but different from

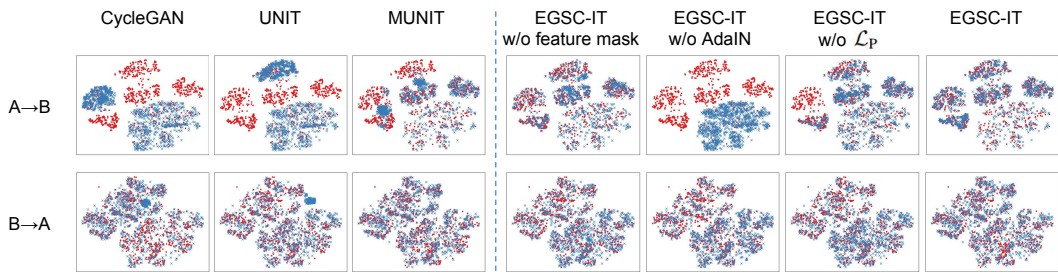

Figure 5: Single-digit translation t-SNE embedding visualization. Red dots: real samples. Blue crosses: generated samples. Best viewed in color.

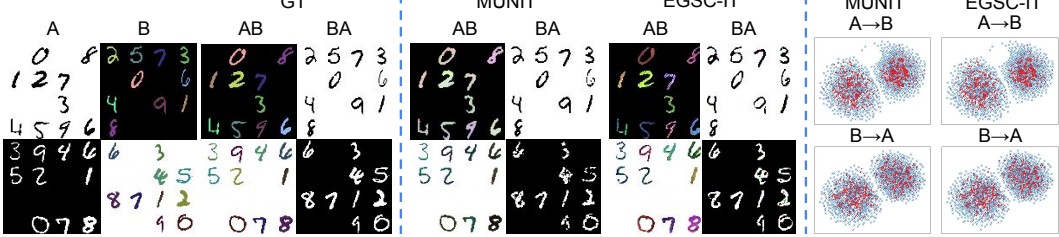

Figure 6: Multi-digit translation. Left: testing results. Right: t-SNE embedding visualization. Red dots: real samples. Blue crosses: generated samples. Best viewed in color.

each other; in domain B, the background is randomly assigned to either *black* or *white* and each foreground digit is assigned to a certain color, but with a little saturation and lightness perturbation. Our goal is to encourage the network to understand the semantic information, *i.e.* the different digits and backgrounds, when translate an image from domain A to domain B. That is, a successfully translated image should have the content of domain A, the digit class, and the style of domain B, the digit and background colors respectively. This experiment is quite challenging, but we observe that our model can still obtain good results without the need for ground-truth semantic labels or paired data. For example, in Figure 6 top row the digits 1,2,3,4,6 can be successfully translated given the criteria described above. As seen in Fig. 6, MUNIT can not translate the foreground color with semantic consistency, and the colors look more "fake".

**Street view translation.** We carry out a synthetic ↔ real experiment for street view translation between GTA5 (Richter et al., 2016) and Berkeley Deep Drive (BDD) (Xu et al., 2017) datasets. The street view datasets are more complex than the digit ones (different illumination/weather conditions, complex environments). As shown in Fig. 7, our method can successfully translate the images from the source to the target domain according to the exemplar's style. For small variations, *e.g.* day→day (first row), MUNIT can keep up, however for large variations, *e.g.* day→night and vice versa (second row), which is exactly the problem we examine in this paper, only EGSC-IT can successfully translate details like the proper sky

Table 3: Semantic segmentation evaluation on 256×512 resolution.

| Method | GTA → BDD | |
|---|---|---|
| | mIoU | mIoU Gap |
| Source | 0.329 | -0.119 |
| UNIT | 0.297 | -0.151 |
| MUNIT | 0.331 | -0.117 |
| EGSC-IT | **0.343** | **-0.105** |
| Oracle | 0.448 | 0 |

color and illumination condition w.r.t. the exemplar. Similar to FCN-score used by Isola et al. (2017), we also use the semantic segmentation performance to quantitatively evaluate the image translation

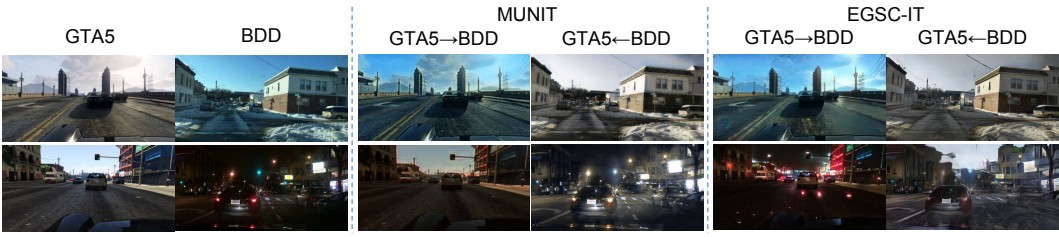

Figure 7: Street view translation testing results. Best viewed in color.

quality. We first translate images in GTA5 dataset to an arbitrary image in BDD dataset. We only generate images of size $256 \times 512$ due to the limitation on GPU memory. Then, we train a single-scale Deeplab model (Chen et al., 2018) on the translated images and test it on BDD test set. The mean Intersection over Union (mIoU) scores in Tab. 3 show that training with our translated synthetic images can improve the segmentation results, which indicates that our method can indeed reasonably translate the source GTA5 image to the target domain style with semantic consistency and reduce the domain difference successfully.

## 5 DISCUSSIONS

Since our method does not use any semantic segmentation labels nor paired data, there are some artifacts in the results for some hard cases. For example, as to the street view translation, day→night and night→day (*e.g.* Fig. 7 bottom row) are more challenging than day→day (*e.g.* Fig. 7 top row). As a result, it is sometimes hard for our model to understand the semantics in such cases. In the future, it would be interesting to extend our method to the semi-supervised setting in order to benefit from the presence of some fully-labeled data.

## 6 CONCLUSION

We introduced the EGSC-IT framework to learn a multimodal mapping across domains in an unsupervised way. Under the guidance of an exemplar from the target domain, we showed how to combine AdaIN with feature masks in order to transfer the style of the exemplar to the source image, while retaining semantic consistency at the same time. Numerous quantitative and qualitative results demonstrate the effectiveness of our method in this particular setting.

## ACKNOWLEDGMENTS

We gratefully acknowledge the support of Toyota Motors Europe, FWO Structure from Semantics project, and KU Leuven GOA project CAMETRON.

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

# A SUPPLEMENTARY MATERIAL

This supplementary material includes more experimental results (§A.1), as well as additional implementation details regarding the network architecture and training (§A.2).

## A.1 EXTRA EXPERIMENT RESULTS

**Face gender translation.** The Large-scale CelebFaces Attributes (CelebA) dataset (Liu et al., 2015) is a large-scale face attributes dataset with more than 200K celebrity images. We divide the aligned face images into male and female domains, containing 84,434 and 118,165 images respectively. We perform face gender translation on this dataset to show how the proposed method can be generalized to tasks with attributes as styles. From Fig. 8, we observe that EGSC-IT can translate the face gender successfully, and most importantly transfer the style of hair, skin and background according to the given exemplar image, unlike MUNIT. In addition, we also provide the male→female face translation results matrix Fig. 9. We observe that the output image's content is consistent with the source image's and its style is consistent with the target image's. Such observation can reflect how the latent space changes given different input images, *i.e.* the the content latent is only related to source image since the style information is combined in the decoder part through feature mask and AdaIN techniques.

**Letter-digit translation.** To further evaluate the generalization ability of our method, we use EMNIST (Cohen et al., 2017) for three translation tasks: 1) black-white letter ↔ colored digit; 2) black-white digit ↔ colored letter; 3) black-white letter ↔ colored letter. EMNIST dataset, a set of handwritten character digits, has the same image format and dataset structure with MNIST dataset. We apply the same process to EMNIST letter images as that for MNIST single-digit images in the main paper. As shown in 10, our model trained with only single-digit data can successfully generalize to the letter data.

**Single-digit translation.** For the quantitative comparison, we also report the SSIM socre of style transfer methods in Tab. 4. In addition, the larger size version of the single-digit translation t-SNE embedding visualization as shown in Fig. 11.

Table 4: SSIM evaluation for single-digit translation. Higher is better. * Models are trained using MNIST-Single data as EGSC-IT.

|  | AdaIN* | WCT* | CycleGAN | UNIT | MUNIT | EGSC-IT |
|---|---|---|---|---|---|---|
| A → B | 0.282±0.103 | 0.061±0.064 | 0.214±0.168 | 0.178±0.160 | 0.463±0.094 | **0.478± 0.090** |
| B → A | 0.188±0.118 | 0.063±0.084 | 0.089±0.166 | 0.074±0.158 | 0.227±0.128 | **0.232± 0.131** |

**Multi-digit translation.** The setting of this experiment was presented in the main paper. Here, we provide more details on the results. As seen in Fig. 12, both CycleGAN and UNIT can not translate the foreground and background color accordingly, and the colors in CycleGAN look more "fake". This is due to the fact that CycleGAN and UNIT only learn a one-to-one mapping. These observations are consistent with the SSIM score in Tab. 5, where both CycleGAN and UNIT have much lower SSIM scores. Differently, MUNIT can not translate the foreground color with semantic consistency, and the colors look more "fake". These observations are consistent with the visualization of t-SNE embeddings. As to the SSIM score, MUNIT seems comparable to ours although visually it is performing worse. The probable reason is that MUNIT can mostly translate the background successfully which occupies the majority of the image.

**Street view translation.** We also provide a larger size version of the results for GTA5 ↔ BDD translation as shown in Fig. 14. In addition, we also provide the ablation study results in Fig. 15. We observe that: 1) removing feature mask will lead to color mismatches or inaccuracies (*e.g.* Fig. 15(a) 1st row 3rd col); 2) removing AdaIN will reduce the model to unimodality (*e.g.* all images are translated to a sunny day with blue sky, see Fig. 15(a) 4th col) since the output image's style is not guided by the exemplar image; 3) removing perceptual loss will lead to incorrect style (*e.g.* Fig. 15(b) 5th col) and the color will spread even given the feature mask since there is no perceptual feedback during training (*e.g.* Fig. 15(a) 5th col).

Table 5: SSIM evaluation for multi-digit translation. Higher is better.

|  | CycleGAN | UNIT | MUNIT | EGSC-IT |
|---|---|---|---|---|
| A → B | 0.145±0.213 | 0.239±0.256 | 0.500±0.035 | **0.503±0.035** |
| B → A | 0.130±0.216 | 0.234±0.257 | **0.501±0.036** | 0.495±0.036 |

## A.2 IMPLEMENTATION DETAILS

The network architecture and training parameters are listed in Tab. 6. We set the number of down-sampling and up-sampling convolutional layers $n_1 = 1$ in single-digit translation and $n_1 = 3$ in other translation experiments. Following UNIT (Liu et al., 2017), the number of residual blocks in $Encoder$ and $Generator$ is set to $n_2 = 4$ with one sharing layer, and the number of convolutional layers in discriminator is set to $n_3 = 5$. The threshold parameter $\eta$ is used to adjust how much the feature mask affects the information flow. Setting $\eta = 0$, *i.e.* using the feature maps (paper Fig. 2 top branch) as feature mask directly, leads to useful information being dropped out and artifacts in the results. Setting $\eta = 1$, *i.e.* not using feature mask at all, leads to results without semantic consistency (see paper Fig. 4). After experimenting with different values, we fixed it to $0.5$. We use the Adam Kingma & Ba (2015) optimizer with $\beta_1 = 0.5$ and $\beta_2 = 0.999$. The learning rate is polynomially decayed with a power of 0.9, as mentioned in (Chen et al., 2018). In order to keep training stable, we update encoder and generator 5 times, and discriminator 1 time in each iteration. The loss weights in $\mathcal{L}_{\text{UNIT}}$ are following (Liu et al., 2017), and $\lambda_c$, $\lambda_s$ are chosen according to the dataset variations and tasks. For data augmentation, we do left-right flip and random crop. In addition, we set a low $\lambda_c$ for face gender translation, since we need to change the shape and add/remove hair in this translation task.

Table 6: Network architecture and training parameters.

| Translation | $n_1$ | $n_2$ | $n_3$ | Minibatch | Learning rate | $\lambda_s$ | $\lambda_c$ | Iteration |
|---|---|---|---|---|---|---|---|---|
| Single-digit | 1 | 4 | 5 | 8 | 1e-5 | 1e3 | 1e1 | ∼60k |
| Multi-digit | 3 | 4 | 5 | 8 | 1e-5 | 1e4 | 1e2 | ∼60k |
| GTA5↔BDD | 3 | 4 | 5 | 3 | 1e-4 | 1e4 | 1e2 | ∼22k |
| Face gender | 3 | 4 | 5 | 8 | 1e-4 | 5e3 | 1e1 | ∼30k |

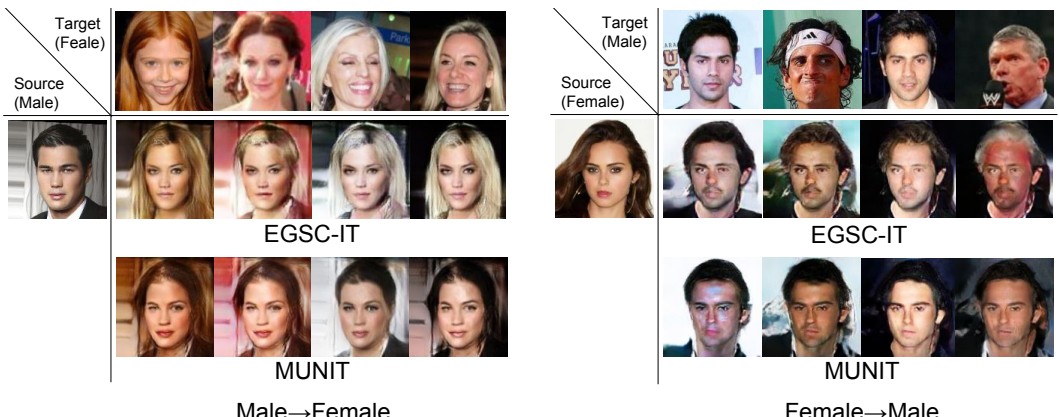

Figure 8: Face gender translation testing results of EGSC-IT. Best viewed in color.

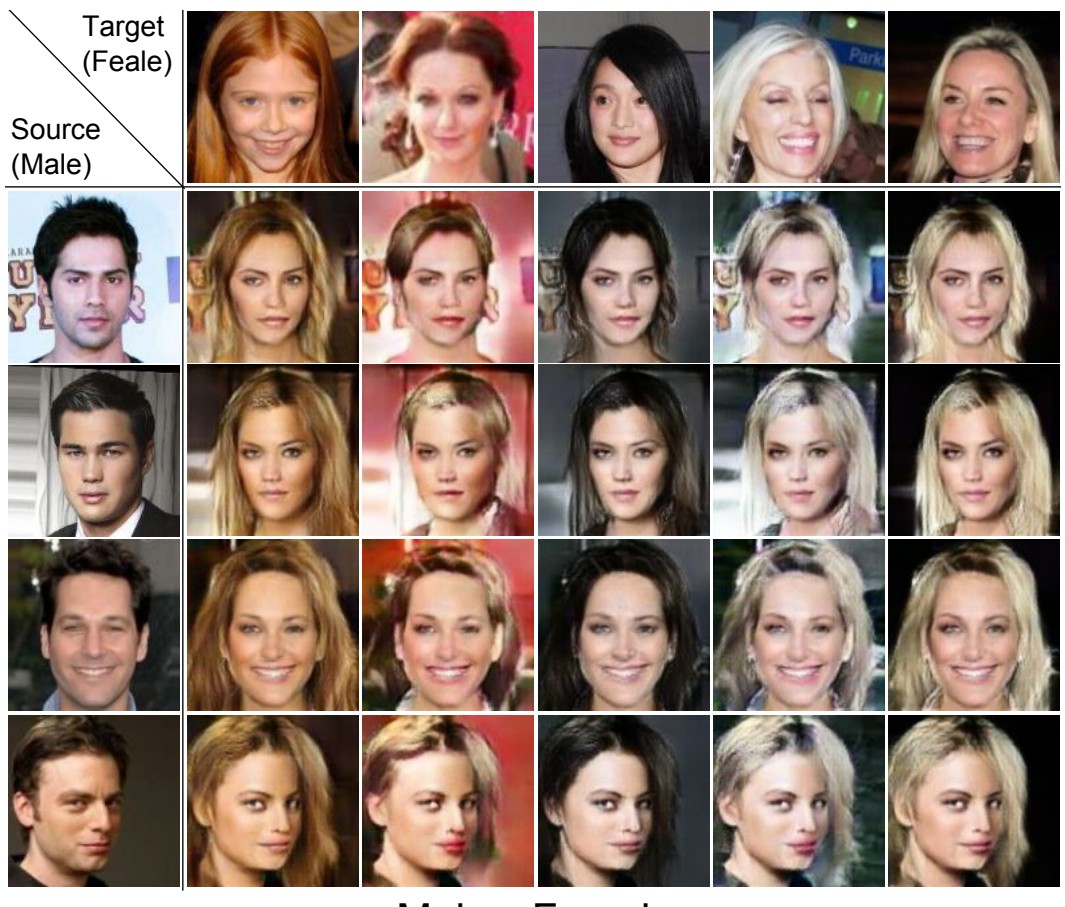

Figure 9: Male→Female translation testing results of EGSC-IT. Best viewed in color.

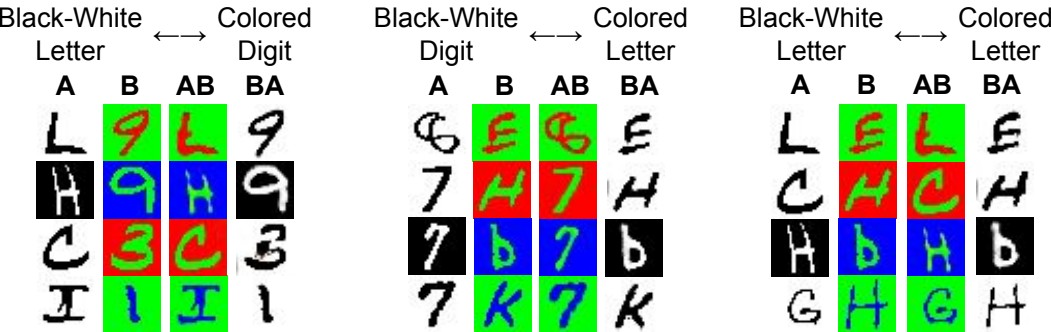

Figure 10: Letter-digit translation testing results. Note that, models are trained only with single-digit data.

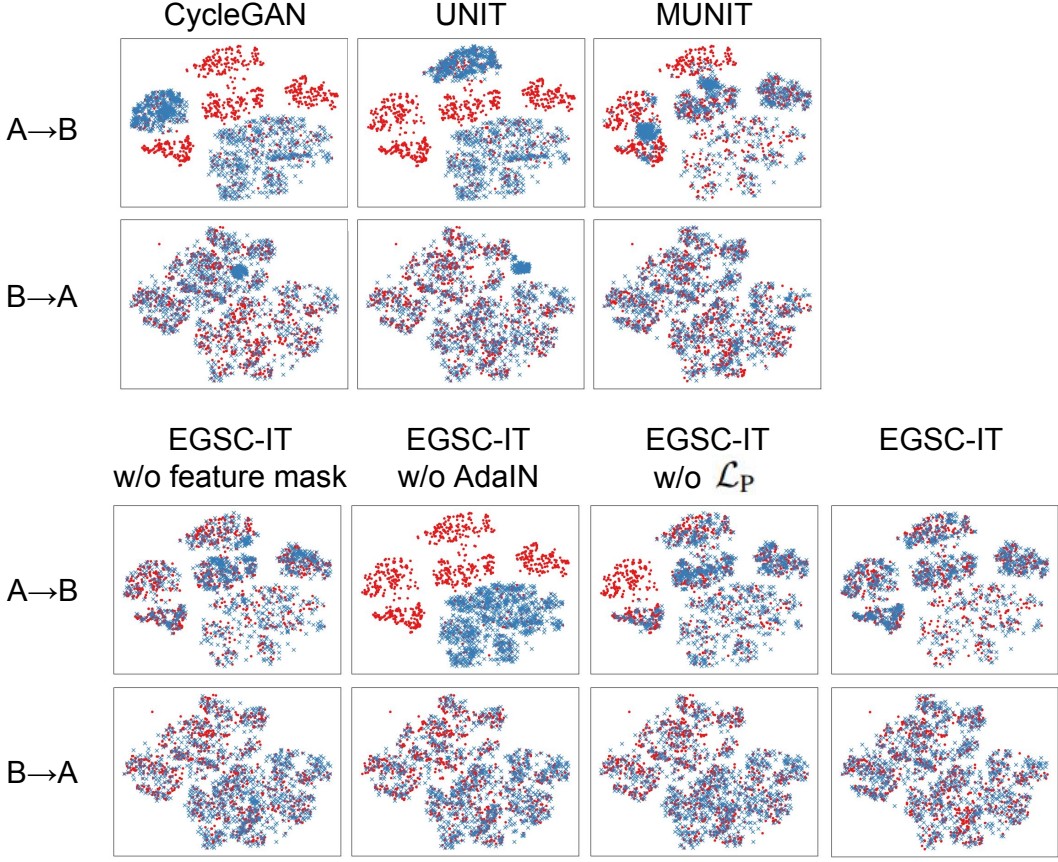

Figure 11: Single-digit translation t-SNE embedding visualization in larger size. Red dots: real samples. Blue crosses: generated samples. Best viewed in color.

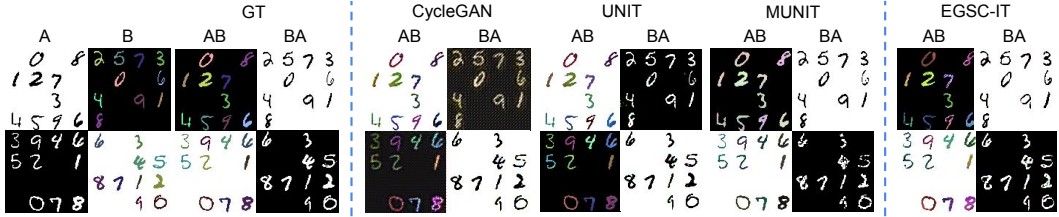

Figure 12: Multi-digit translation testing results.

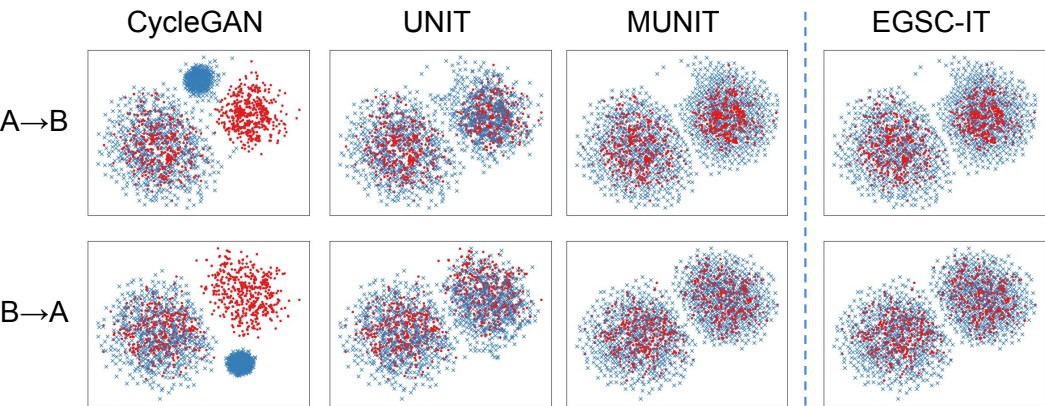

Figure 13: Multi-digit translation t-SNE embedding visualization. Red: real samples. Blue: generated samples. Best viewed in color.

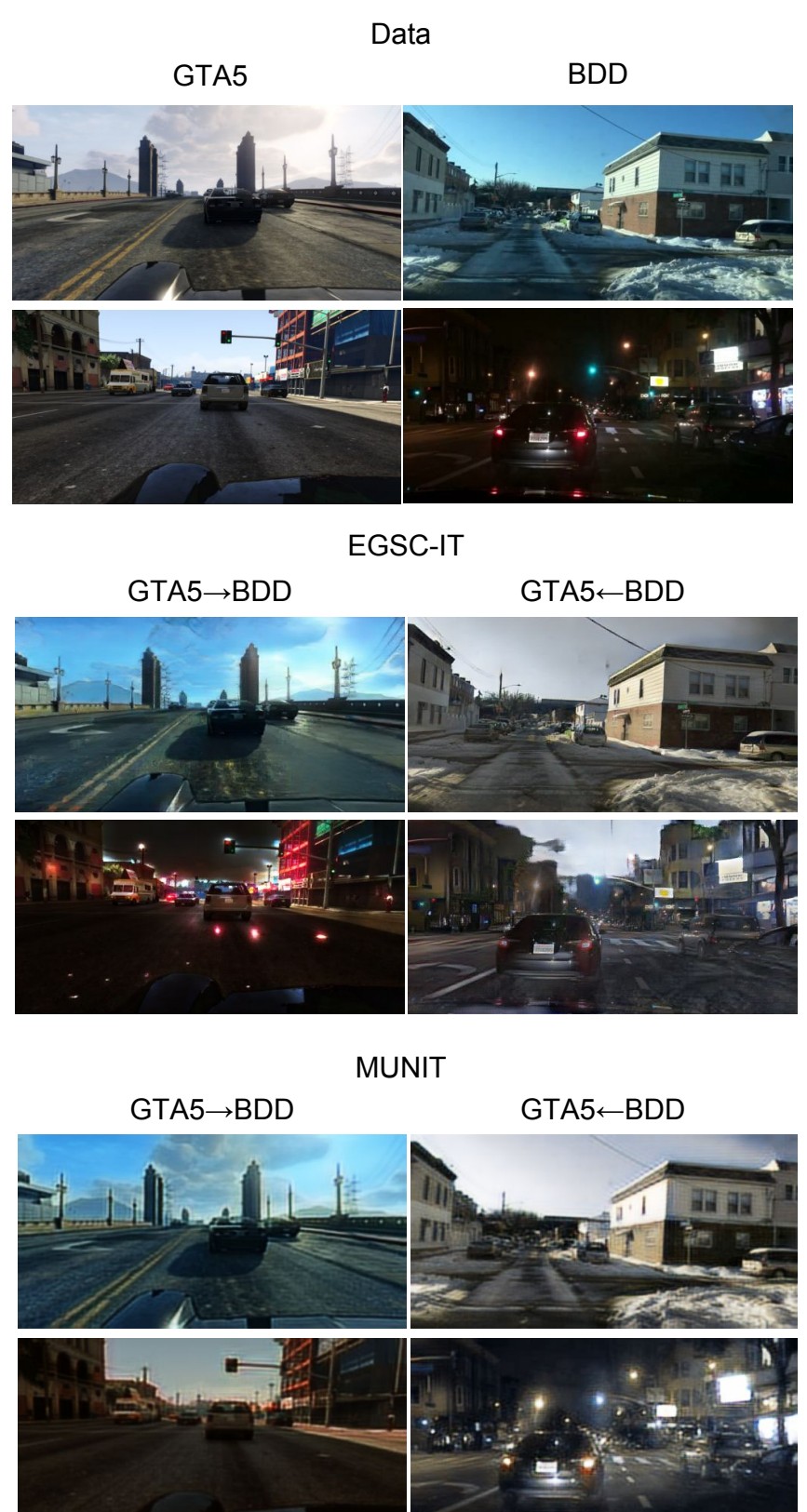

Figure 14: Street view translation testing results in larger size. Best viewed in color.

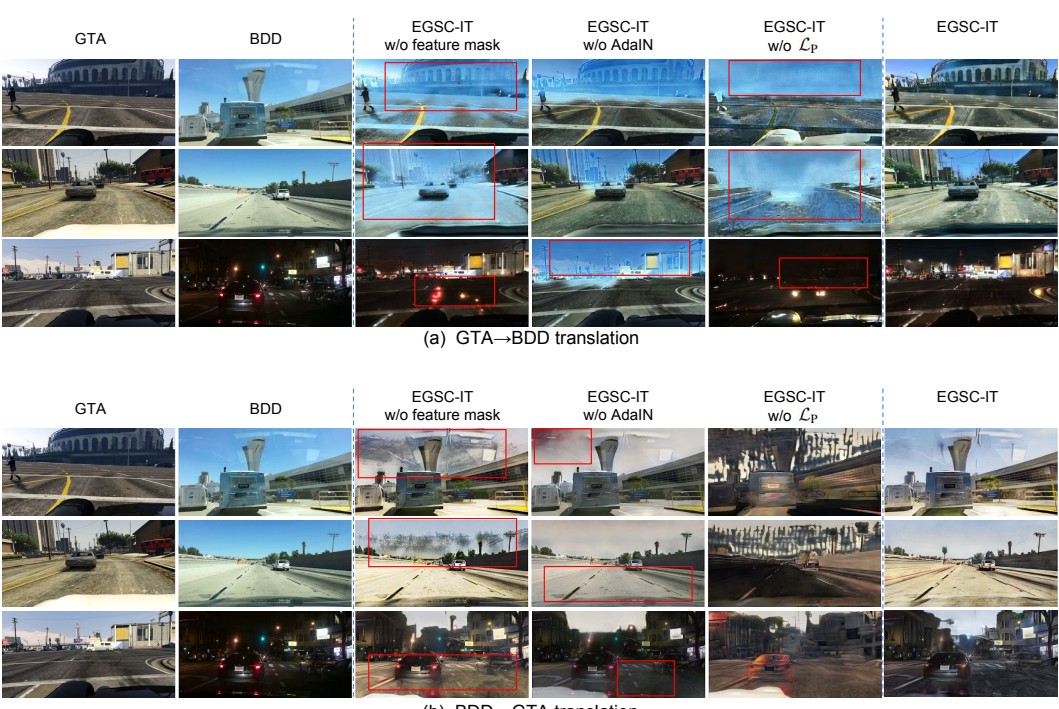

Figure 15: Street view translation ablation study results. Zoom in for more details. Best viewed in color.

