# OpenReview forum: "Exemplar Guided Unsupervised Image-to-Image Translation with Semantic Consistency"
_ICLR.cc/2019/Conference_

### Official Review · AnonReviewer3 · 2018-10-30
**interesting submission with good intuition and evaluation**

**Rating:** 8
**Confidence:** 4

**Review:**

I enjoyed reading this manuscript. The paper is based on a simple idea used by others as well (i.e., the image has two components, one  that encode content which is shared across domains and another one characterizing the domain specific style). The other important idea is the use of feature masks that steer the translation process without requiring semantic labels. This is similar to attention models used by others but I think it is novel when applying to this specific application domain. I was a bit disappointed by the evaluation part. The authors decided to perform ablation and to show the importance of each component using only the MNIST-Single dataset. While this is good as a toy example I would have expected to see such analysis on a more complex example, e.g., street-view translation. This is also surprising considering that it is not even present in the supplementary material. Overall, this is a solid submission with interesting ideas and good implementation.

---

> ### Author Response · Authors · 2018-11-23
> **Response to AnonReviewer3**
>
> We thank Reviewer 3 for the constructive review and detailed comments.
>
> 1. Ablation study
> In our paper, we present the ablation study on the MNIST-Single dataset because it is a more controlled setting where we can generate ground truth for comparisons. Furthermore, as mentioned in a previous answer, we believe that this simplified experiment offers a more direct and intuitive way to evaluate the expected translations compared to e.g. street-view translation. However, as Reviewer 3 suggested, it is interesting to show the ablation study results on more complex examples too. As such, we added these results in the supplementary material in Fig. 15. We observe that: 1) removing the feature mask leads to color mismatches or inaccuracies (e.g. Fig. 15(a) 1st row 3rd col); 2) removing AdaIN reduces the model to unimodality (e.g. all images are translated to a sunny day with blue sky, see Fig. 15(a) 4th col) since the output image's style is not guided by the exemplar image; 3) removing perceptual loss leads to incorrect style (e.g.  Fig. 15(b) 5th col) and the color  spreads even given the feature mask since there is no perceptual feedback during training (e.g.  Fig. 15(a) 5th col).

---

### Official Review · AnonReviewer2 · 2018-11-02
**Justifying style transfer via conditioning needs more analysis.**

**Rating:** 5
**Confidence:** 5

**Review:**

The introduction is written to perfection. The paper discusses a core failing and need for I2I translation models. The one-to-one mapping assumption does not apply to most tasks. While the approach seems novel the analysis of the results are insufficient to convince me that the method is really working. This should be a workshop paper.

For the motivation of the approach I am not convinced how the conditioned style is being used. It would be nice to see some analysis of how the latent space changes given different input images. Why would style information be propagated through the network? Why wouldn't noise work just as well? Although an abiliation study is performed there is no standard deviation reported so it is unclear if this number is fair.

In Figure 5 the t-sne doesn't look correct. The points all seems to be projected on walls which could indicate some sort of overflow error. The text only devotes 3 lines to discuss this figure. It is not mentioned what part of the model the t-sne is computed from. To me this experiment that studies the internal representation is critical to convincing a reader to use this method.

The segmentation results sound good. Where is the improvement coming from? The experimental section is cut short. The experiment section is really squeezed in the last two pages while the other sections are overly descriptive and could be reduced.

The figures should be changed to be visible without color (put a texture on each block).

---

> ### Author Response · Authors · 2018-11-23
> **(2 | 2) Response to AnonReviewer2**
>
>
> 3. t-SNE embedding visualization and standard deviation of SSIM scores.
> The t-SNE embeddings are calculated from the translated images. We first use PCA to reduce the dimension to 50. Then, we use the t-SNE implementation in the sklearn package to compute the t-SNE embeddings. We have tried different t-SNE parameters and choose the good ones for visualization. That is to say, the t-SNE figures are generated using standard techniques w.r.t related works. Previously, we set 'init=pca' for discriminative dense point clouds. As suggested, we now set 'init=random' for a better visualization of the distributions. This results in nicer visualizations, although the difference between some of the methods becomes less outspoken. Reviewer 2 asked to modify some t-SNE parameters to avoid the "projected on wall" visualization. We updated the figures for better visualizations, but the tendency is the same. A larger size version of the single-digit translation t-SNE embedding visualization is shown in Fig. 11. We also changed the markers from '1','2' to '.','x' in the t-SNE figures for better visualization without color. The standard deviation of SSIM scores are added to Tab. 2 and Tab. 5 as requested.
>
> 4. Segmentation improvement.
> The segmentation improvement is the natural outcome of trying to preserve semantic consistency during the translation process. Our translated GTA->BDD images are semantically consistent w.r.t. the original GTA images, i.e. the sky is still the sky and trees are still trees, which allows us to obtain improved segmentation performance as e.g. the semantic boundaries are better delineated in the generated images compared to techniques that do not account for semantic consistency. Another advantage is that our model can handle large within domain variations, such as day and night. As a result, when using the translated images with paired GTA semantic segmentation labels to train a segmentation model, the domain difference will be reduced, and as such the segmentation results will also improve.
>
>
> [b] Xun Huang and Serge J. Belongie. Arbitrary style transfer in real-time with adaptive instance normalization. In ICCV, 2017.
>
> [c] Leon A Gatys, Alexander S Ecker, and Matthias Bethge. Image style transfer using convolutional neural networks. In CVPR, 2016.
>
> [d] Jun-Yan Zhu, Richard Zhang, Deepak Pathak, Trevor Darrell, Alexei A Efros, Oliver Wang, and Eli Shechtman. Toward multimodal image-to-image translation. In NIPS, 2017.

---

> ### Author Response · Authors · 2018-11-23
> **(1 | 2) Response to AnonReviewer2**
>
> We thank Reviewer 2 for the constructive review and detailed comments. We apologize if the experimental section text due to space constraints feels short and leads to misunderstandings. We are going to clarify everything below. Please see the blue fonts in the newly uploaded draft to check how our paper is changed, to be in accordance with the following responses.
>
>
> 1. Style information propagation.
> As already mentioned, the style information in our method is propagated by using the Adaptive Instance Normalization (AdaIN) technique [b]. This is a well-known technique in the style transfer field which has proven to be very successful for arbitrary style transfers, and has been adopted by many follow-up works. The idea behind AdaIN in our case is to align the mean and variance of the content feature channels coming from domain A (after applying the feature mask) with those of the style feature channels coming from domain B. According to [c], these feature statistics have been found to carry the style information in an image. Noise can also be used to generate images with diverse style [d], as proposed by Reviewer 2. However, our goal in this work is to allow users more explicit control over the translation process, which is something that noise-based approaches do not allow. In particular, noise inputs do not easily translate to intuitive style guidance, in contrast to our exemplar-guided approach, where we propose to use a sub-network (F_B in Fig. 2) to explicitly extract the feature statistics from the exemplar image itself and - through AdaIN - adapt accordingly the source image. As a result, the user can match any desired style from the target domain just by picking the corresponding exemplar.
>
> 2. Latent space.
> With respect to the previous question, Reviewer 2 asks for a visualization of how the latent space changes given different exemplar images. Although generally a valid request, it does not apply to our case. Let us explain why. Due to the specifics of our architecture, the latent space is only associated with the content representation of an image, not the style of the exemplar. The latter is only added after the encoder part, i.e. the latent space, and the feature mask sub-network through the AdaIN sub-network (see Fig. 2). As such, changing exemplars and visualizing the latent space would give the same point in the latent space if the source image (that provides the content) is the same. Alternatively, changing source images and visualizing the latent space is not informative about the translation procedure as the style is only added after the feature mask and AdaIN sub-networks. In summary, the latent space is only related to the source image since the style information is only combined in the decoder part through feature mask and AdaIN techniques. Instead, to mimic what Reviewer 2 asked for, we added the male->female face translation results matrix in Fig. 9. We observe that the output image's content is consistent with the source image and its style is consistent with the target image. Such observation can reflect how the latent space changes given different source images as well as exemplars.

---

### Official Review · AnonReviewer1 · 2018-11-03
**Interesting and well-written paper, but needs some clarification on the experiments**

**Rating:** 6
**Confidence:** 4

**Review:**

The paper is well organized with a clear idea of the proposed method and good related work descriptions. Overall, the descriptions are clear and easy to follow, but the experimental results need clarifying.

- Regarding the multi-digit translation task, it is not straightforward to this reviewer how the proposed method could match the digits (semantic) with different colors (style) in different locations. The description in the paper is not enough to explain the results in Fig. 6. To this reviewer, this task is more complex than the street view translation one. In the same line, it is curious what the results would be if digits with different colors are overlapping at random location, rather than the grid-like arrangement. 

- For the potential readers who are not knowledgeable in semantic segmentation, please give the full name of mIoU for reference.

- For further researches in this topic, it would be good to depict the limitations of the proposed method. For examples, the translated images in the CelebA dataset are not photorealistic (Fig. 8)  and there are odd red lights in the middle of the results in GTA5<-BDD (Fig. 12).

- typos: Fig. 2-caption: m_{a}->m_{A}

---

> ### Author Response · Authors · 2018-11-23
> **Response to AnonReviewer1**
>
> We thank Reviewer 1 for the constructive review and detailed comments. Below, we respond to each comment in detail. Please see the blue fonts in the newly uploaded draft to check how the paper has changed, to be in accordance with the following responses.
>
>
> 1. Multi-digit translation.
> We added more explanation for the multi-digit translation experiment to the paper.
>
> (1) The MNIST-Multiple dataset is a controlled experiment, similar to MNIST-CD/CB [a], designed to verify our method's ability to disentangle content and style representations during I2I translation. In particular, the content is represented by the different digits (0-9) and background classes, whereas style is represented by the shape and color variation added in digits as well as the (black or white) background colors. Our goal is to encourage the network to understand the semantic information, i.e. the different digits and backgrounds, when translating an image from domain A to domain B. That is, a successfully translated image should have the content of domain A, i.e. the digit class, and the style of domain B, i.e. the digit and background colors respectively.
>
> (2) We can achieve this by using the proposed EGSC-IT model. First, the feature mask sub-network (F_A in Fig. 2) extracts the content information, i.e. the digit class, and provides it to the decoder. Second, the employed perceptual loss encourages the decoder to learn how to use this content information to do the translation while retaining semantic consistency. Finally, the AdaIN sub-network (F_B in Fig. 2) translates the style information of each semantic class, i.e. the digit and background colors respectively, by using AdaIN, which has proven very successful in arbitrary style transfer tasks (see later responses for a further comment on this). We refer to the theoretical part of the paper where we provide a very detailed description of the translation procedure.
>
> (3) We agree with Reviewer 1 that this experiment is quite challenging, but we observe that our model can still obtain good results without the need for ground-truth semantic labels or paired data. For example, in Figure 6 top row the digits 1,2,3,4,6 can be successfully translated given the criteria described above. In street view translation the scenes in an image are generally more complex - i.e. the within class variation in the BDD dataset is much larger than that of the synthetic MNIST-Multiple dataset. We designed this experiment as a controlled, yet challenging, task to evaluate the different image-to-image translation methods. Given the lack of ground-truth translated images to compare to in our setting - due to the unsupervised nature of our problem - we believe this simplified experiment offers a more direct and intuitive way to evaluate the expected translations compared to e.g. street-view translation.
>
> (4) The setting of different colors that overlap at random locations, as proposed by Reviewer 1, seems very interesting in theory. However, we believe that in practice it would be very difficult for any unsupervised translation method as there is too much ambiguity in the overlapped locations for a network to decide where to draw style cues from.
>
> 2. Full name of mIoU.
> We added the full name 'mean Intersection over Union' (mIoU). Thank you for the useful note.
>
> 3. Limitations.
> Since our method does not use any semantic segmentation labels nor paired data, there are still some artifacts in the generated images for some hard cases. This seems natural given the difficulty of the task. For example: (a) in street view translation, day->night and night->day (e.g. Fig. 7 bottom row) are more challenging than day->day (e.g. Fig. 7 top row). As a result, it is sometimes hard for our model to understand the semantics in such cases. Even state-of-the-art fully-supervised semantic segmentation networks suffer in low light or adverse weather conditions. (b) in face gender translation, our model can successfully translate the gender attribute while keeping the semantics, e.g. skin, hair and background color, consistent with the exemplar image. However, since we do not provide any semantic segmentation labels this results in some artifacts. This discussion about limitations will be added in the paper. In the future it would be interesting to extend our method to the semi-supervised setting in order to benefit from the presence of some fully-labeled data.
>
> 4. Typos.
> We fixed them. Thank you for finding them.
>
>
> [a] A. Gonzalez-Garcia, J. van de Weijer, Y. Bengio. Image-to-image translation for cross-domain disentanglement. In NIPS, 2018.

---

### Author Response · Authors · 2018-11-23
**Summary of the first revision**

We thank all reviewers for their constructive reviews and detailed comments. We are committed to incorporate all the proposals to further improve the original draft. Below, we respond to each comment in detail. In the first revision, we provide additional experimental results in Appendix A and more explanations in both the main paper and Appendix A. Please see the blue fonts in the newly uploaded revision to check how the paper has changed according to your indications. Note that, any future suggestions or requests from the reviewers will also be incorporated similarly to the blue font changes.

We strongly believe that we made an important and novel step towards solving the unsupervised image-to-image translation problem.

If you have any further questions or suggestions, please do not hesitate to let us know.

Many thanks again for all your sincere contributions on ICLR 2019,
The authors.

---

### Meta-Review · Area_Chair1 · 2018-12-13
**New technique for semantic consistency for transfer across heterogenous domains with preliminary empirical evidence**

**Confidence:** 4
**Recommendation:** Accept (Poster)

**Metareview:**

This paper proposes an image to image translation technique which decomposes into style and content transfer using a semantic consistency loss to encourage corresponding semantics (using feature masks) before and after translation. Performance is evaluated on a set of MNIST variants as well as from simulated to real world driving imagery.

All reviewers found this paper well written with clear contribution compared to related work by focusing on the problem when one-to-one mappings are not available across two domains which also have multimodal content or sub-style.

The main weakness as discussed by the reviewers relates to the experiments and whether or not the set provided does effectively validate the proposed approach. The authors argue their use of MNIST as a toy problem but with full control to clearly validate their approach. Their semantic segmentation experiment shows modest performance improvement. Based on the experiments as is and the relative novelty of the proposed approach, the AC recommends poster and encourages the authors to extend their analysis of the current results in a final version.